# From Discovery to Innovative Translational Approaches in 80 Years of Fragile X Syndrome Research

**DOI:** 10.3390/biomedicines13040805

**Published:** 2025-03-27

**Authors:** Mathijs B. van der Lei, R. Frank Kooy

**Affiliations:** Center of Medical Genetics, University of Antwerp and Antwerp University Hospital, 2650 Edegem, Belgium; mathijs.vanderlei@uantwerpen.be

**Keywords:** fragile X syndrome (FXS), trinucleotide repeat expansion (CGG repeat), fragile X messenger ribonucleoprotein 1 (*FMR1*), fragile X messenger ribonucleoprotein (FMRP)

## Abstract

Fragile X syndrome (FXS) is the most common inherited cause of intellectual disability and a major genetic contributor to autism spectrum disorder. It is caused by a CGG trinucleotide repeat expansion in the *FMR1* gene, resulting in gene silencing and the loss of FMRP, an RNA-binding protein essential for synaptic plasticity. This review covers over 80 years of FXS research, highlighting key milestones, clinical features, genetic and molecular mechanisms, the FXS mouse model, disrupted molecular pathways, and current therapeutic strategies. Additionally, we discuss recent advances including AI-driven combination therapies, CRISPR-based gene editing, and antisense oligonucleotides (ASOs) therapies. Despite these scientific breakthroughs, translating preclinical findings into effective clinical treatments remains challenging. Clinical trials have faced several difficulties, including patient heterogeneity, inconsistent outcome measures, and variable therapeutic responses. Standardized preclinical testing protocols and refined clinical trial designs are required to overcome these challenges. The development of FXS-specific biomarkers could also improve the precision of treatment assessments. Ultimately, future therapies will need to combine pharmacological and behavioral interventions tailored to individual needs. While significant challenges remain, ongoing research continues to offer hope for transformative breakthroughs that could significantly improve the quality of life for individuals with FXS and their families.

## 1. Introduction

Fragile X syndrome (FXS) is a neurodevelopmental disorder characterized by a wide range of cognitive, behavioral, and medical challenges. It is the most common inherited cause of intellectual disability (ID) and the leading single-gene form of autism spectrum disorder (ASD), affecting approximately 1 in 4000 males and 1 in 8000 females [1,2,3]. This review offers a comprehensive overview, beginning with a timeline of significant milestones and key events that have advanced our understanding of FXS over the past 80 years. Subsequently, we examine the clinical phenotype, underlying genetic and molecular mechanisms, the FXS mouse model, the distinct affected pathways, and current therapeutic strategies. Finally, we discuss novel, innovative, and exciting approaches, including AI-driven combination therapies, CRISPR-based gene editing, and antisense oligonucleotide (ASO) treatments, which are paving the way for transformative advancements in FXS research.

## 2. From Discovery to Translation: 80 Years of Progress in FXS Research

FXS was first documented over 80 years ago, with significant advancements and milestones since its discovery (Figure 1). In 1943, British geneticists J.P. Martin and J. Bell conducted an extensive family case study, identifying 11 males across two generations with what was characterized as ‘imbecility’ [4]. This terribly outdated term was later replaced by “mental retardation” and more recently revised to “intellectual disability” to align with contemporary diagnostic and clinical nomenclature [5]. Martin and Bell suggested that the condition was sex-linked, heritable, and associated with the incomplete development of specific brain regions. Although they did not fully understand the cause, the disorder was initially named Martin–Bell syndrome.

In 1969, Lubs and colleagues provided further insight and clarity into the genetics of FXS [6]. They studied three generations of a family with four males affected by intellectual disability and performed karyotyping to examine their chromosomes. Karyotyping, which assesses the size, shape, and number of chromosomes in a cell sample, revealed a breakage near the end of the long arm of the X chromosome at position Xq27.3, known as the fragile site FRAXA (fragile site, X chromosome, A). In 1981, seven members of the original family described by Martin and Bell were re-examined, and five were found to carry the same fragile site on the X chromosome [7]. This case report also noted, for the first time, the typical facial features and macroorchidism (abnormally large testes) associated with FXS.

The following decades showed significant advancements in recombinant DNA technologies, which are techniques to isolate, clone, and identify DNA segments of interest. These developments enabled the identification of disease genes. After extensive research across multiple labs, a trinucleotide repeat expansion (CGG repeat) within the fragile X messenger ribonucleoprotein 1 (*FMR1*) gene was identified in 1991 [8,9,10]. Normally, this DNA segment is approximately repeated 5 to 55 times, but in individuals with FXS, it is repeated over 200 times [11]. This abnormal expansion leads to the methylation of the *FMR1* gene, silencing it and preventing the production of fragile X messenger ribonucleoprotein (FMRP), resulting in FXS.

The lack of brain material from FXS patients made it challenging to study the mechanisms underlying the disorder. To accelerate research and gain more insight into the pathological and physiological processes, animal models were developed. Over 30 years ago, in July 1994, the first *Fmr1* knockout (KO) mouse model was created by the Dutch-Belgian Fragile X Consortium [12]. This model, along with the later developed conditional *Fmr1* KO [13], recapitulates key features of FXS, including the absence of FMRP, macroorchidism, learning deficits, and hyperactivity [14,15,16]. Over subsequent decades, extensive research using the *Fmr1* KO model identified deficits in receptors, signaling pathways, and downstream targets, advancing our understanding of FXS pathophysiology and driving the development of potential therapeutics [17,18,19,20,21,22,23]. These advancements also attracted investment from major pharmaceutical companies, spurred the establishment of the first FXS research foundations, and raised public awareness, including the creation of National Fragile X Awareness Day and dedicated Fragile X research centers of excellence in the USA [24,25,26,27].

Despite these efforts, therapeutic success in preclinical models has rarely translated into clinical trials, underscoring the complexity of the disorder. This has led to the exploration of innovative strategies, such as AI-driven combination therapies, which aim to target multiple pathways simultaneously for a more holistic treatment approach [28]. Additionally, promising advances in *FMR1* gene targeting therapies offer new avenues for reactivating the *FMR1* gene and hold significant potential for future breakthroughs [29,30]. These exciting approaches, along with their challenges and implications, will be discussed in detail later in this review.

## 3. Understanding the Clinical Phenotype of FXS: Cognitive and Physical Manifestations

FXS is characterized by a broad spectrum of clinical manifestations affecting cognitive, behavioral, physical, and other domains (Table 1). These clinical features vary in severity among affected individuals depending on the extent of FMRP deficiency and other genetic and environmental factors. Understanding the full range of clinical features is crucial for accurate diagnoses, management, and therapeutic interventions for FXS.

Cognitive impairment is a core feature of FXS, with affected individuals exhibiting varying degrees of intellectual disability. Typically, IQ scores fall below 70, with males averaging 40–55 and females 65–70, though considerable variability exists [23,31]. Affected individuals exhibit deficits in working and short-term memory, executive function, language, and verbal memory, along with impairments in adaptive skills essential for communication (such as planning), self-care, and social participation [31,32,33]. Individuals with FXS frequently experience developmental delays, including motor and language delays, alongside learning impairments [1].

Beyond cognitive deficits, individuals with FXS exhibit distinct behavioral phenotypes, including hyperactivity, anxiety, aggression, attention deficits, sensory hyper arousal, and stereotypic behaviors such as hand flapping and unusual speech patterns like echolalia. Hyperactivity and impulsivity are particularly prominent in boys and often meet the criteria for attention-deficit/hyperactivity disorder (ADHD). Anxiety disorders are also highly prevalent, with social anxiety being a common issue [1,32,34]. FXS shares many features with autism spectrum disorders (ASD), characterized by social communication deficits and restricted and repetitive behaviors that emerge in early childhood and impair daily functioning [33]. Approximately 30–50% of males and 20% of females with FXS exhibit autistic behaviors by direct assessment, such as repetitive movements, gaze avoidance (limited eye contact), sensory sensitivities, and social communication difficulties [32,34]. This phenotypic overlap underscores the need to consider FXS in the differential diagnosis of ASD, particularly in cases with a family history of intellectual disability.

Individuals with FXS also exhibit distinct physical features, though these may be subtle in infancy and become more pronounced with age. Characteristic traits include an elongated face, large ears, and a prominent jaw and forehead [1,34,35]. Additional physical characteristics include hyperflexible joints, particularly in the fingers, and flat feet. Post-puberty males with FXS may exhibit macroorchidism, a specific physical marker of FXS [1,34].

Finally, FXS individuals are prone to various comorbid health conditions, such as recurrent otitis media (ear infection), strabismus (the eyes do not line up with each other when looking at an object), and weight gain [36]. Seizures are relatively common, occurring in approximately 15–20% of males and 5% of females with FXS and are generally well-managed with medication, underscoring the need for neurological monitoring [37,38]. Gastrointestinal problems, including gastroesophageal reflux, constipation, and irritable bowel syndrome, are common and may contribute to behavioral disturbances [23]. Sleep disturbances, such as difficulty falling asleep, frequent awakenings, and restless sleep, further highlight the need for targeted interventions to improve sleep hygiene and quality [39].

## 4. Genetic Basis of FXS: The Role of the *FMR1* Gene and FMRP Protein

The genetic mutation responsible for FXS involves an expansion of an unstable CGG trinucleotide repeat within the 5′ untranslated region (UTR) of the *FMR1* gene [8,9,40]. This gene is located on the X chromosome at position Xq27.3 and comprises 17 exons [41], with a widespread expression, including in spermatogonia, and is particularly abundant in neurons [42,43,44]. The total number of CGG repeats divides the population into three distinct categories (Figure 2).

In the general population, the number of CGG repeats ranges from approximately 5 to 55, considered as “the normal healthy range” [45]. The premutation (55–200 repeats) leads to elevated *FMR1* mRNA and a moderate reduction in FMRP levels but does not typically cause FXS. However, it is associated with related disorders such as fragile X-associated tremor/ataxia syndrome (FXTAS) and fragile X-associated primary ovarian insufficiency (FXPOI) [46,47]. FXTAS predominantly occurs in older males with premutation alleles [48], though female carriers can also be affected [49]. Affected individuals exhibit various neuropathologies, resulting in clinical symptoms such as intention tremors and cerebellar gait ataxia, which are areas of ongoing research for potential treatments [50]. Premutation alleles also predispose females to FXPOI, characterized by early menopause, reduced fertility, low bone density, earlier onset of coronary heart disease, anxiety, and depression [51]. Finally, the full mutation, defined as over 200 CGG repeats, induces hypermethylation and silencing of the *FMR1* gene. This epigenetic modification prevents the transcription of *FMR1* mRNA, thereby eliminating the production of FMRP, leading to FXS [52,53]. The exact mechanisms driving the CGG repeat expansion remain incompletely understood, but the formation of unusual secondary DNA structures at expanded repeats is likely to contribute to instability and is currently being investigated using cellular and animal models (as reviewed in [54]).

FMRP is an RNA-binding protein widely expressed in the brain, particularly in regions associated with learning and memory, such as the hippocampus and cerebral cortex. The protein is also found at the bases of neuronal dendrites and within dendritic spines [42,55,56]. FMRP binds to several mRNAs, controlling their transport and translation at synapses. This regulation is critical for synaptic maturation and plasticity, ensuring that proteins necessary for synaptic development and remodeling are synthesized at the right time and place, which is crucial for normal neuronal development and functioning [57,58,59]. FMRP regulates the synthesis of many synaptic proteins through its functional domains, including three K homology (KH) domains (KH0, KH1, and KH2), which are essential for binding mRNAs, especially those involved in synaptic protein synthesis [60,61]. Additionally, an arginine-glycine-glycine (RGG) box plays a key role in RNA-protein interactions, influencing RNA processing, splicing, stability, and transport, along with other protein-interaction domains [62]. The protein contains a nuclear localization signal (NLS) and a nuclear export signal (NES), enabling it to shuttle between the cytoplasm and nucleus [43,61,63]. At the N-terminus, two amino-terminal Agenet domains, also known as the tandem Tudor domain, may bind trimethylated lysines, which regulate protein stability and function [61,64] (Figure 2).

One of the key consequences of FMRP deficiency is the abnormal development of dendritic spines, which are small protrusions on the dendrites of neurons where synapses form [65]. In FXS, these spines are often immature and abnormally elongated, with impaired synaptic connectivity [66,67]. This dysmorphogenesis contributes to widespread neural circuit dysfunction, affecting both local synaptic interactions and long-range brain connectivity. The resulting network disorganization underlies the cognitive and behavioral deficits in FXS, including intellectual disability, autistic traits, and social anxiety [68]. Thus, the loss of FMRP-driven translational regulation creates an imbalance in synaptic protein synthesis, which ultimately impairs neural connectivity and communication.

Finally, loss of FMRP disrupts multiple signaling pathways crucial for synaptic function. For instance, FMRP negatively regulates the excitatory metabotropic glutamate receptor 5 (mGluR5) pathway, which is involved in synaptic plasticity and long-term depression (LTD) [17,18]. This dysregulation extends to other neurotransmitter systems, including specific components of the inhibitory γ-Aminobutyric acid (GABAergic) system [69,70,71,72,73,74]. The resulting imbalance between excitatory and inhibitory signaling leads to hyperexcitability and increased neural activity in FXS. Additionally, as discussed later in this review, many other pathways and targets contribute to FXS.

**Figure 2 biomedicines-13-00805-f002:**
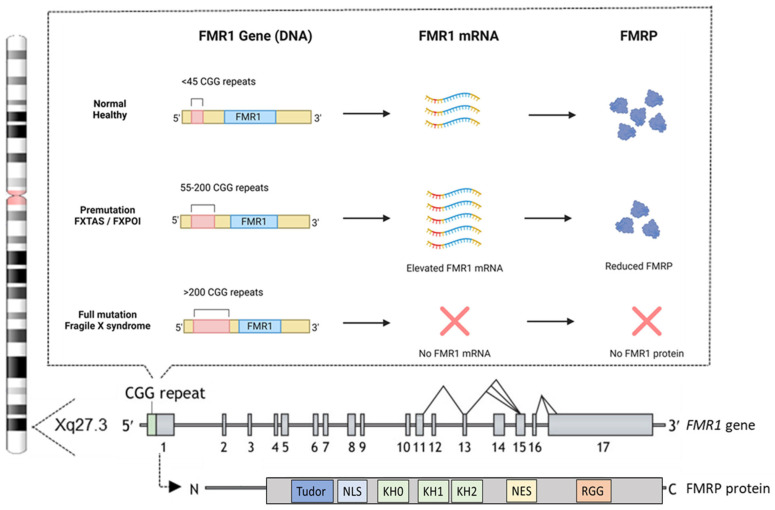
CGG repeats and the structure of the *FMR1* gene and FMRP protein. Most individuals have fewer than 45 CGG repeats in their *FMR1* gene, which is considered normal and results in typical *FMR1* mRNA levels and FMRP production. Individuals with 55–200 CGG repeats carry the premutation, which is associated with elevated *FMR1* mRNA levels and a moderate reduction in FMRP production. Those with more than 200 CGG repeats have the full mutation, where *FMR1* transcription is silenced due to promoter hypermethylation, leading to the absence of FMRP production. At the bottom is a schematic overview of the human *FMR1* gene, highlighting its potential alternative splicing sites, along with the human FMRP protein and its functional domains. These domains include a Tudor methyl-lysine- and methyl-arginine-binding domain, a nuclear localization signal (NLS), three K homology (KH) domains, a nuclear export signal (NES), and an arginine-glycine-rich (RGG) domain. The figure is partially adapted from [75] and created with BioRender.com.

## 5. Modeling FXS in Mice: Insights into the Pathophysiology of FXS

Over the last three decades, multiple genetically modified animal models have been generated for FXS. The *FMR1* gene is highly conserved through evolution, which makes it possible to study the gene in other organisms, too [8]. The creation of these models was essential, as no naturally occurring models existed. These models enable researchers to investigate *Fmr1* function and explore potential therapies. This review focuses on the *Fmr1* KO mouse model, the most widely used and validated preclinical model for FXS [12]. Other models, including conditional *Fmr1* KO mice, transgenic overexpression models, and permutation mouse models, have been generated (as reviewed in [75]). Additionally, a rat model has been developed for FXS, and even in non-mammalian models like the fruit fly *Drosophila melanogaster* and the zebrafish *Danio rerio* [16,76,77].

The first *Fmr1* KO mouse model was generated by the Dutch-Belgium Fragile X Consortium in 1994 and remains in use today [47]. This model was created by replacing the wild-type murine *Fmr1* gene with a nonfunctional version through homologous recombination. The insertion of a neomycin resistance cassette into exon 5 results in the loss of intact *Fmr1* mRNA and FMRP production, although small amounts of mutated RNA are transcribed [12]. The phenotypic features of the *Fmr1* KO mouse model closely resemble those seen in FXS, supporting its validity. Notably, *Fmr1* KO mice exhibit skull alterations, akin to the distinctive facial features observed in FXS, and consistently show macroorchidism, a hallmark of FXS in post-pubertal males [12,35].

FXS individuals also have a higher density of immature cortical dendritic spines, a neuroanatomical abnormality also observed in the *Fmr1* KO mouse model [66,78,79,80,81]. These mice show increased protein synthesis in brain homogenates and synaptoneurosomes, implicating the role of FMRP in synaptic plasticity [82,83]. Furthermore, *Fmr1* KO mice display reduced long-term potentiation (LTP) and elevated long-term depression (LTD), which are mechanisms underlying learning and memory [84,85]. LTP strengthens the connections between the presynaptic and postsynaptic parts of the neuron, enhancing signal transduction sensitivity, while LTD weakens these connections. A balance between LTP and LTD is essential for cognitive processes, with FMRP regulating this equilibrium.

Behaviorally, *Fmr1* KO mice exhibit phenotypes compatible with symptoms observed in FXS individuals. Over the past 30 years, various behavioral tests have demonstrated that *Fmr1* KO mice display mild cognitive impairments in tasks like the Morris water maze, although results vary depending on genetic background [12,86,87,88]. These mice also show increased locomotor (hyper)activity, altered acoustic startle response, prepulse inhibition, anxiety, and social dominance, reflecting common FXS traits [14,89,90,91]. While spontaneous seizures are rare, they can be induced by audiogenic stimuli, with susceptibility varying by genetic background [72,92].

Mice are, of course, quite different from humans; they are much smaller, live shorter lives, and learn less over their lifespan. Another fundamental difference is that *Fmr1* KO mice never produce FMRP from birth, although the *Fmr1* promoter remains intact, leading to residual transcription of abnormal *Fmr1* RNA at around 27% of wildtype levels [12,13], whereas FXS individuals have active *FMR1* expression until at least the 10th week of gestation [93,94]. Despite these differences, the mouse model is well-accepted because mice share almost 99% of their genes with humans and exhibit similar physiological and pathogenic mechanisms [95]. Lastly, mice are relatively inexpensive to house, practical to breed, and have short generation times and lifespans, making them an ideal and valuable model for FXS research.

## 6. Disrupted Pathways in FXS: A Complex Network of Affected Pathways and Targets

The mouse model has significantly enhanced our understanding of the pathophysiology of FXS. Preclinical studies over the past decades have demonstrated the critical role of FMRP in multiple cellular pathways, with several potential therapeutics showing efficacy in reversing symptoms. The role of FMRP in the excitatory glutamatergic system and the inhibitory GABAergic system is especially highlighted in relation to a potential excitatory/inhibitory neurotransmission imbalance associated with FXS [18,20]. Furthermore, the absence of FMRP has been linked to deficits in the endocannabinoid system, muscarinic acetylcholine receptors, and several intracellular signaling pathways, such as extracellular signal-related kinase (ERK1/2) [96], phosphodiesterase 4 (PDE4) [97,98,99], phosphatidylinositol 3-kinase (PI3K) [100,101], and downstream targets, such as matrix metallopeptidase-9 (MMP9) [102,103,104], mammalian target of rapamycin (mTOR) [105,106], p70 ribosomal S6 kinase 1 (S6K1) [107], diacylglycerol kinase kappa (DGKκ) [108], glycogen synthase kinase 3 (GSK3) [109,110], eukaryotic translation initiation factor 4E-binding protein (4E-BP) [111], and p21-activated kinases (PAK) [112] (reviewed in [113,114,115]). Collectively, multiple receptors, intracellular signaling pathways, and downstream targets are affected in FXS, underscoring the complexity of the molecular basis of FXS and the challenges in developing effective treatments.

While many promising findings have emerged from mouse models, these often fail to translate to success in clinical trials, leading to the saying, “*It’s the best time to be an FXS mouse because everything seems to work*”. Nevertheless, hope persists as ongoing clinical trials explore targeted treatments across multiple pathways. (Table 2). As of March 2025, a search on ClinicalTrials.gov for recruiting interventional studies specifically for FXS highlights several pharmacological approaches focused on (1) GABA pathways deficits, (2) glutamatergic N-methyl-D-aspartate (NMDA) receptor dysfunction, (3) targeting reduced cyclic adenosine monophosphate (cAMP) levels with predominantly phosphodiesterase-4D (PDE4D) inhibitors, (4) activation of AMP-activated protein kinase (AMPK), and (5) modulation of the endocannabinoid (EC) system. Finally, two studies targeting calcium-activated potassium big potassium (BK) channels have shown promising results. These different therapeutic approaches are discussed in detail in the following sections.

### 6.1. Modulating the Inhibitory GABAergic System in FXS: GABA_A_ and GABA_B_ Receptors

The GABAergic system is the main inhibitory system in the brain, involved in various functions including anxiety, epilepsy, circadian rhythm, and processes related to learning and memory [73,74]. There are two main types of receptors: ionotropic GABA_A_ receptors and the metabotropic G-protein coupled GABA_B_ receptors. The GABA_A_ receptors consist of 19 known subunits, and activation of the receptor by GABA binding results in chloride influx, which mediates postsynaptic membrane hyperpolarization in mature neurons to reduce neuronal excitability and firing [116]. Multiple molecular and electrophysiological studies have shown a reduction in the expression of specific GABA_A_ receptor subunit expression and a decrease in GABA_A_ receptor-mediated inhibition in FXS [19,69,71,74,117,118]. Individuals with FXS exhibit reduced GABA_A_ receptor availability [119], paralleling the observations made in the FXS mouse model.

Two GABA_A_ receptor agonists have been tested preclinically in the FXS mouse model and even in individuals with FXS. Ganaxolone, the first of these, showed promising results in the mouse model, rescuing audiogenic seizures and reducing stereotypic and repetitive behaviors [70,72]. This success in preclinical studies led to a clinical trial involving nearly 60 children aged 6 to 17 with FXS [120]. Although no significant benefits were observed in the primary and secondary outcome measures, a post hoc analysis revealed significant improvements in anxiety and hyperactivity scores in the most anxious subgroup of children compared to placebo [120]. Additionally, a second subgroup of children with the lowest IQ showed improvements in the same domains [120]. Despite the overall study not showing significant improvements in the primary outcomes, these subgroup analyses suggest potential benefits in specific populations within FXS.

Gaboxadol, also known as THIP or OV101, is another GABA_A_ receptor agonist. In preclinical studies, gaboxadol rescued neuronal hyperexcitability [117] and significantly reduced hyperactivity and prepulse inhibition [121]. Recent studies further demonstrated that gaboxadol restored various aberrant behaviors, including hyperactivity, anxiety, irritability, aggression, and repetitive behavior, although it did not improve cognitive functions [28,122]. In clinical trials, gaboxadol was found to be safe and well tolerated in individuals with FXS. A 12-week randomized double-blind study showed an initial efficacy signal, with improvements based on clinician- and caregiver-rated endpoints that assessed behaviors such as hyperactivity, irritability, stereotypy, and anxiety [123]. Currently, they are recruiting adult males with FXS for a single-dose challenge study of gaboxadol (NCT06334419).

In addition to the GABA_A_ receptor, the GABA_B_ receptor has been a target of drug therapy in FXS. The GABA_B_ receptors consist of the GABA_B1_ and GABA_B2_ subunits. Presynaptic GABA_B_ receptors inhibit GABA release by reducing calcium influx via voltage-gated Ca^2+^ channels, while postsynaptic GABA_B_ receptors induce hyperpolarization by the activating of potassium (K^+^) channels [124]. The GABA_B_ receptor agonist arbaclofen (also known as STX209) has shown promising results in preclinical studies, and reversed audiogenic seizures, excess protein synthesis, abnormal spine density, and repetitive and social behaviors after administration in mice [125,126,127,128].

The promising preclinical data on arbaclofen led to two Phase 3 placebo-controlled clinical trials in adults and children with FXS. The adult trial did not show significant efficacy on the primary outcome measures [129]. However, in the pediatric trial involving children aged 5 to 11, there were beneficial effects observed in irritability subscales and the parent-scored assessment forms, although the primary and other secondary outcome measures were not significant [129]. Recently, a new entity acquired the rights to arbaclofen and plans to conduct further studies in children with FXS. It is hoped that these future studies will identify a subgroup of individuals who respond positively to the treatment. Notably, many families have reported anecdotally that arbaclofen was the most effective medication their child had ever used.

### 6.2. The Excitatory Glutamatergic System in FXS: From Potential Cure to Failure

Glutamate is the main excitatory neurotransmitter in the brain and a key treatment target in FXS. Aberrant signaling through metabotropic glutamate receptors (mGluRs), particularly the mGluR5 receptor, plays a crucial role in the pathophysiology of FXS. The “mGluR theory” proposed by Bear and colleagues suggests that the absence of FMRP leads to enhanced glutamatergic signaling via mGluR5 receptors, which leads to increased protein synthesis and altered synaptic plasticity, including enhanced LTD [17,18]. Several mGluR negative allosteric modulators, mainly directly targeting the mGluR5 receptor, have been developed and tested in the mouse model of FXS. Studies investigating these negative allosteric modulators, such as 2-methyl-6-(phenylethynyl) pyridine (MPEP), showed the rescue of synaptic plasticity and cognitive deficits in a Drosophila model of FXS [130]. Multiple cellular, electrophysiological, and behavioral phenotypes of the FXS mouse model are also rescued after acute treatment. These rescues include the abnormal prepulse inhibition, increased anxiety and repetitive behaviors, increased seizure susceptibility, and aberrant spine morphology [131,132,133].

However, MPEP exhibits low specificity, leading to its replacement by 2-chloro-4-((2,5-dimethyl-1-(4-(trifluoromethoxy)phenyl)-1H-imidazol-4-yl)ethynyl)pyridine (CTEP), a more selective mGluR5 negative allosteric modulator with a much longer duration of action. Acute CTEP treatment has been shown to reduce the incidence of audiogenic seizures, correct deficits in LTD, and lower elevated protein synthesis [134]. These findings underscore the role of mGluR5 in the pathogenesis of FXS. Although, a crucial question remains unanswered: can pharmacological mGluR5 inhibition reverse an already established FXS phenotype in the mouse model? To address this, another study initiated a 30-day chronic treatment with CTEP in mice aged 4–5 weeks of age, a developmental stage where the brain is anatomically mature but still highly plastic. This chronic treatment rescued cognitive deficits, auditory hypersensitivity, and aberrant spine density. Additionally, it corrected excessive protein synthesis and ameliorated macroorchidism [134]. Collectively, these studies suggest that targeting the mGluR5 receptor, even in young adulthood, could be a highly effective and promising therapy for individuals with FXS.

These encouraging and promising animal studies led to the broad interest of major pharmaceutical companies, leading them to invest in the development of mGluR5 inhibitors for treating FXS. Mavoglurant (also known as AFQ056), a non-competitive mGluR5 inhibitor developed by Novartis, demonstrated the rescue of several phenotypes in the mouse model of FXS [135,136,137]. However, it failed to show statistically significant differences from placebo in primary caregiver-rated outcomes in clinical trials [138,139]. Similarly, Roche developed basimglurant, another mGluR5 negative allosteric modulator, which also did not demonstrate efficacious effects in adolescents or adults with FXS [140]. Questions remain regarding the design and execution of these trials. Concerns include whether the trials used optimal primary endpoints, whether the age of the participants (adolescents or adults instead of young children) was appropriate, and whether most studies were statistically underpowered. Additionally, the trials have been prone to large placebo effects as they were the first targeted treatments for FXS, and families had strong expectations of their success.

Nevertheless, a recent clinical trial was designed to address these concerns and was the first large, multisite study focused on 3- to 6-year-old children with FXS, specifically examining the effects on learning [141]. Each patient was studied over 12 months, starting with a multi-month placebo lead-in, followed by 2 months of dose optimization to the maximum tolerated dose, and 6 months of treatment using a learning paradigm known to be effective in FXS. Although both groups made language progress and no safety issues were reported, there were no significant differences between the AFQ056 and placebo-treated groups [141]. Thus, despite the preclinical evidence in the FXS mouse model supporting the efficacy of negative allosteric modulators targeting mGluR5, the mechanism proposed by the “mGluR theory” did not translate into observable benefits for individuals with FXS.

As previously discussed, much of the drug development effort for FXS has focused on attenuating the excessive activation of mGluR5. However, dysregulated activity of ionotropic glutamate receptors may also play a role in FXS. These ionotropic receptors, which are ligand-gated ion channels, mediate the majority of excitatory synaptic transmission and are key players in synaptic plasticity [142]. They are categorized into several functional subtypes: α-amino-3-hydroxy-5-methyl-4-isoxazolepropionic acid (AMPA) receptors, kainate receptors, N-methyl-D-aspartate (NMDA) receptors, and GluD receptors [142]. An imbalance between AMPA and NMDA receptor activity, along with inappropriate internalization of these receptors, has been implicated in the excessive LTD observed in FXS [17,143]. Memantine, an NMDA receptor antagonist, has shown promise in a small open-label trial involving individuals with FXS, where improvements were reported [144]. Currently, a Phase 2 clinical trial of memantine is underway at the Cincinnati Children’s Hospital Medical Center, representing the only study specifically targeting glutamate receptors in FXS (NCT05418049).

### 6.3. Phosphodiesterase-4D (PDE4D) in FXS: From Discovery in 1990s to Phase 3 Clinical Trials

Alterations in cyclic AMP (cAMP) metabolism are consistently observed in both individuals with FXS and animal models of the condition. As early as the 1990s, a defect in cAMP production was reported in FXS, resulting in decreased cAMP levels [145,146,147]. Since then, it has been proposed that inhibiting phosphodiesterase-4 (PDE4) could prevent this cAMP degradation [146,147,148]. This alteration in cAMP metabolism has also been demonstrated in the fruit fly and mouse model of FXS, where decreased cAMP levels were observed in the brain [97,98,99]. Roflumilast was the first PDE4 inhibitor but lacked specific selectivity for the D subunit. This specific PDE4D plays a crucial role in modulating cAMP levels relevant to human cognition, as ultra-rare autosomal dominant mutations in PDE4D have been linked to a neurodevelopmental syndrome with intellectual disability [149]. The most promising PDE4D-specific negative allosteric modulator is BPN14770 (also known as Zatolmilast). This compound ameliorated various behavioral phenotypes in the FXS mouse model, such as hyperactivity and decreased social interaction, while also improving natural behaviors like nest building and marble burying [98]. Surprisingly, this study alone was sufficient to initiate the transition to human clinical trials.

The effects of BPN14770 were evaluated in a randomized, placebo-controlled study involving 30 adults with FXS aged 18 to 41 years. The study demonstrated that the compound was well tolerated, with no adverse side effects reported [150]. Notably, the treated individuals with FXS exhibited significant improvements across a broad range of outcome measures, including behavior, quality of life, and most importantly, cognition [150]. This was the first study to demonstrate cognitive improvements in adults with FXS. A recent Phase 2 study involving 30 individuals with FXS has been completed, although the results have yet to be published (NCT03569631). Currently, three additional Phase 3 clinical trials are underway, each recruiting a minimum of 150 participants. These trials target male FXS adolescents aged 9 to 18 years (NCT05163808) and male adults aged 18 to 45 years (NCT05358886), respectively. Participants who complete any of these clinical trials have the option to enroll in a two-year open-label extension study to further assess the long-term safety and tolerability of BPN14770 (NCT05367960).

### 6.4. Exploring the Potential of CBD Transdermal Gel in FXS

Cannabidiol (CBD) is the non-psychotropic component of cannabis, while delta-9-tetrahydrocannabinol (THC) is the psychotropic component [151]. CBD interacts with the endocannabinoid (EC) system, a key modulator of synaptic plasticity, cognitive performance, anxiety, nociception, and seizure susceptibility [152]. In the absence of FMRP, the EC system becomes dysregulated and is unable to maintain the balance between inhibitory and excitatory neurotransmitter release, potentially leading to some of the phenotypic characteristics of FXS [151,153,154]. ZYN002, a transdermal gel containing 4.2% (*w*/*w*) cannabidiol, has been evaluated in clinical trials for its efficacy in FXS. A Phase I/II open-label trial demonstrated significant improvements in anxiety, hyperactivity, aggression, and ADHD symptoms in children with FXS, without significant side effects [155]. However, a subsequent international, multicenter, double-blind, randomized, controlled Phase III clinical trial in children and adolescents with FXS did not achieve statistical significance for the primary endpoint in the full cohort but showed significant improvements in patients with >90% *FMR1* methylation [156].

To further investigate these findings, a new study with 250 FXS individuals is being conducted at multiple centers, focusing on individuals with FXS aged 3–30 years who have full mutation and full methylation of the *FMR1* gene (NCT04977986).

### 6.5. Metformin: Can the Repurposed “Wonder Drug” Also Benefit FXS?

Metformin is an FDA-approved compound primarily used to lower blood glucose levels in patients with type 2 diabetes. In addition to its glucose-lowering effects, metformin has demonstrated efficacy and safety in treating obesity in both children and adults, irrespective of their diabetic status [157,158,159]. The drug also seems to have beneficial effects on (the prevention of) cardiovascular disease, cancer, and dementia [160]. Beyond these effects of metformin, recent studies in FXS animal models have shown that metformin can improve cognitive and behavioral phenotypes. Specifically, in the FXS fruit fly, metformin rescued memory deficits [161], while in the FXS mouse model, metformin ameliorated core deficits [162]. The underlying mechanisms by which metformin exerts these effects remain complex. However, it is known that metformin lowers activity in the mechanistic target of rapamycin (mTOR) and mitogen-activated protein kinase (MEK)-extracellular signal-regulated kinase (ERK) pathways, both of which are upregulated in FXS [163]. This reduction is mediated through the activation of AMP-activated protein kinase (AMPK), which enhances mitochondrial function and consequently downregulates the mTOR and MEK-ERK pathways. Additionally, metformin’s ability to decrease blood glucose levels leads to a downregulation of the insulin receptor, which is also upregulated in FXS [161,163].

These promising preclinical results have prompted the exploration of metformin as a treatment for individuals with FXS. Clinical studies have shown that metformin is well tolerated, with patients aged 4 to 60 years exhibiting positive behavioral changes, including reduced irritability, social avoidance, and aggression, as well as benefits in appetite and weight control after at least six months of treatment [164]. Similar positive outcomes were observed in a trial involving young children aged 2 to 7 years, where general improvements in development, language, and behavior were reported [165]. Furthermore, a case study involving two adult FXS patients treated with metformin for one year showed improvements in IQ scores and behavior, suggesting a potential protective effect against cognitive decline [166]. In recent years, a multicenter controlled trial has been conducted in the United States and Canada, involving FXS patients aged 6 to 25 years. The results of this trial are eagerly awaited (NCT03479476, NCT03722290). Currently, two additional clinical trials are recruiting FXS individuals: one targeting patients aged 2 to 16 years (NCT05120505) and another involving individuals aged 6 to 35 years (NCT03862950). Metformin appears to be a strong candidate for a new targeted treatment for FXS, but the results of the controlled trial need to clarify efficacy.

### 6.6. Unlocking New Possibilities: BK Channel Modulators in FXS

Big potassium (BK) channels are large-conducted Ca^2+^-activated potassium (K^+^) channels that play a crucial role in numerous brain processes. These channels facilitate the flow of large amounts of K^+^ across the cell membrane when open, a function essential for the proper functioning of synaptic connections [167]. FMRP is known to regulate the open time of BK channels in excitatory neurons, thereby influencing neurotransmitter release and synaptic transmission [168,169]. Genetic upregulation of BK channels has been shown to correct many deficits in the mouse model of FXS, suggesting that FMRP directly modulates BK channel activity [169,170]. In February 2025, Spinogenix (San Diego, CA, USA), a clinical-stage biopharmaceutical company, announced promising topline results from a Phase 2 clinical trial of SPG601, a small-molecule therapy targeting BK channels to correct synaptic dysfunction in FXS (NCT06413537) [171]. The randomized, double-blind, placebo-controlled crossover study in 10 adult males with FXS demonstrated that SPG601 significantly reduced high-frequency gamma band activity, a well-established neurophysiological marker of FXS linked to impaired learning and memory. The FDA granted SPG601 Orphan Drug status in May 2024 and Fast Track designation in December 2024, while Spinogenix and Cincinnati Children’s Hospital are completing analysis of the full study results in preparation for publication.

Similarly, Kaerus Bioscience (London, UK) has initiated the development of a more specific BK channel modulator, KER-0192, and successfully completed a Phase 1 clinical trial [172]. The study demonstrated that KER-0193 was safe, well-tolerated, and exhibited dose-proportional pharmacokinetics in 56 healthy volunteers. A biomarker substudy using electroencephalography (EEG) confirmed significant pharmacodynamic effects, providing clinical evidence of central target engagement and demonstrating a region-specific impact on brain excitability that aligns with EEG abnormalities in FXS. These findings establish proof of the mechanism, supporting further development. Kaerus is now preparing for a Phase 2 proof-of-concept trial in FXS patients. Thus, the mechanism of BK channel openers involves increasing the open time of these channels, which reduces neuronal excitability and may alleviate symptoms of FXS. Furthermore, BK channels are implicated in various neurological disorders beyond FXS, including Parkinson’s and Alzheimer’s diseases, highlighting the vast therapeutic potential of this class of drugs [173].

## 7. Novel Innovative Approaches: Combination and Gene Therapies

Recent challenges in FXS research have catalyzed the exploration of novel strategies, marking an exciting era for the field. Despite significant advances in understanding FXS molecular mechanisms, translating these insights into effective treatments remains challenging. To address this, several innovative and multidisciplinary approaches are currently in various stages of development and are discussed. AI-driven combination therapies are leveraging machine learning to identify synergistic drug combinations that target multiple pathways. Advances in understanding the demethylation of *FMR1* and other gene-editing technologies have opened possibilities for directly correcting or reactivating the gene. ASO treatments, in particular, show great promise to reactivate the silenced *FMR1* gene with high specificity. These novel strategies are increasingly being integrated with precision medicine approaches to enhance the likelihood of therapeutic efficacy but also minimize potential side effects.

### 7.1. AI-Driven Combination Therapies Targeting Multiple Distinct Affected Pathways in FXS

Given FMRP’s involvement in numerous distinct pathways, early treatment approaches that target individual pathways have proven overly simplistic, thereby hindering successful clinical translation. This realization has led to the development of combination therapies aimed at addressing the disorder’s complexity by targeting multiple distinct pathways simultaneously. Additionally, advances in AI and big data are transforming the field, with numerous companies leveraging extensive public databases to train algorithms capable of predicting novel therapeutic targets and compounds.

A recent study highlights the potential of data-driven approaches in FXS drug discovery. Using a data-driven computational drug discovery platform, Healx (Cambridge, UK) predicted that combining a GABA_A_ receptor agonist (gaboxadol) with a PDE4/10 inhibitor (ibudilast) would rescue more phenotypes than either drug alone [28,174]. This prediction was generated using Healx’s Combination Gene Expression Matching (CGEM) and Target Optimisation (TargOpt) platforms. Consistent with these predictions, the combination therapy outperformed monotherapies in the FXS mouse model, underscoring the promise of combination treatments [28].

Another example is Kantify (Brussels, Belgium), which leverages its AI-driven SAPIAN platform to identify multiple novel therapeutic targets for FXS and other diseases [175]. Non-AI-driven combination therapies have also shown potential. For example, a small Phase 2 open-label trial of lovastatin and minocycline demonstrated safety, improved behavioral outcomes, and reduced cortical excitability in FXS individuals [176]. However, anticipated synergistic effects are not always guaranteed, as seen in studies targeting both mGluR5 and GABAergic pathways, which failed to enhance outcomes and slightly worsened social behaviors in the FXS mouse model [177].

Despite their promise, developing combination therapies with FDA-approved drugs remains challenging. Additional data are required to demonstrate the safety and efficacy of different combinations, and study design complexities must be addressed. Moreover, if the drugs originate from different manufacturers, intellectual property and collaboration agreements may further complicate development and commercialization.

### 7.2. Adeno-Associated Virus (AAV) Vectors to Restore FMR1 Function

Gene therapy strategies to restore *FMR1* function are emerging as promising approaches for FXS treatment. Adeno-associated virus (AAV)-based therapies gained significant attention for their potential to effectively address the underlying molecular deficits in FXS [178,179]. Multiple studies have demonstrated partial restoration of FMRP, reversing biochemical and physiological abnormalities and attenuating behavioral deficits in FXS mouse and rat models [180,181,182,183,184]. However, these studies used nonhuman FMRP driven by promoters distinct from the human *FMR1* promoter. A recent study using the human *FMR1* promoter successfully rescued several deficits in social behavior, stereotypic and repetitive behavior, as well as dendritic abnormalities, in FXS mice without affecting WT controls [185].

Another example is using an AAV vector to deliver a modified diacylglycerol kinase κ (DGK-κ), a key mRNA target of FMRP that regulates lipid signaling, which is downregulated in *Fmr1* KO mice [108,186]. This delivery via an AAV vector corrected lipid signaling imbalances and improved FXS-relevant behaviors in the knockout model, highlighting its therapeutic potential [187]. However, further research is needed to clarify the role of DGK-κ in FXS pathogenesis and assess its viability as a therapeutic target. Nevertheless, AAV-based gene therapy remains a viable avenue for FXS treatment.

### 7.3. R-Loop Driven Reactivation of FMR1 in FXS

Lee and colleagues developed a strategy to reactivate *FMR1* by targeting the formation of R-loops (three-stranded nucleic acid structures) within extended CGG repeats [30]. A combination of MEK and BRAF inhibitors in cellular models of FXS can strongly reactivate *FMR1* expression and also trigger a significant contraction of the expanded *CGG* repeats. The study uncovered a self-reinforcing (positive feedback) cycle: demethylation of the *FMR1* promoter leads to renewed de novo *FMR1* transcription, which promotes R-loop formation. These R-loops, in turn, recruit the cell’s endogenous DNA repair mechanisms, which precisely remove the excess CGG repeats. Importantly, this repeat contraction is highly specific to *FMR1*, ensuring that other regions of the genome remain unaffected. As a result, the normal expression of *FMR1* is restored, leading to the production of functional FMRP. Overall, this strategy not only reactivates the *FMR1* gene but also corrects the underlying genetic mutation. While further studies are needed to validate this approach in ex vivo and in vivo models, it offers a compelling potential method toward a curative therapy for FXS.

### 7.4. Advances in CRISPR-Based Reactivation of FMR1 in FXS

Recent advances in the CRISPR-based genome and epigenome editing have provided promising strategies for reactivating *FMR1* expression. Three key studies worked on distinct yet complementary approaches: triplet repeat editing, repeat deletion, and targeted DNA demethylation. The first study used CRISPR/Cas9-mediated editing to shorten the expanded CGG repeats in *FMR1* using iPSC-derived neurons from FXS patients [188]. By trimming the CGG repeat length below the pathological threshold, a decreased promoter methylation and corresponding reactivation of *FMR1* transcription were observed. This reactivation restored functional FMRP expression and rescued neuronal phenotypes, including synaptic connectivity and electrophysiological properties, demonstrating that precise repeat editing can reverse key cellular deficits in FXS.

Another study used a complementary approach, where CRISPR/Cas9 technology was used to excise the entire expanded CGG repeat region [189]. The complete deletion of the repeat expansion led to a robust reversal of *FMR1* promoter hypermethylation. Unlike the partial editing strategy in the first study, this approach resulted in a more consistent and durable restoration of *FMR1* expression, as the methylation trigger (the expanded repeats) was entirely removed. The restored FMRP levels normalized synaptic function and neuronal electrophysiology, suggesting the potential for recovery.

The third study used a different method than the previous two genome-editing approaches, and focused on directly reversing the epigenetic silencing of *FMR1* without altering the underlying CGG repeat expansion [190]. The researchers utilized a CRISPR-dCas9 (deactivated Cas9) system fused with TET1, a DNA demethylase enzyme, to selectively remove aberrant methylation at the *FMR1* promoter. This demethylation reactivated *FMR1* transcription and restored FMRP expression, leading to partial recovery of neuronal function in FXS-derived neurons and sustained *FMR1* expression in vivo after neuronal transplantation. Notably, this strategy preserved the genetic integrity of the *FMR1* locus, avoiding the potential risks associated with double-strand breaks and large-scale genomic modifications. This approach offers a safer, reversible alternative that can be fine-tuned to achieve optimal reactivation. The key challenge remains ensuring long-term stability of the demethylated state, as endogenous methylation machinery may restore the silenced epigenetic state over time. Future research should focus on further in vivo validation, long-term safety, and stability to advance CRISPR-based therapies for FXS.

### 7.5. ASO Therapy for FXS

A recent study by the Richter lab revealed that over 70% of individuals with FXS transcribe the *FMR1* gene in white blood cells. However, the *FMR1* RNA is mis-spliced to produce a little-known abnormal splice isoform, *FMR1-217*, which includes exon 1 and a pseudo-exon from intron 1, but cannot produce functional FMRP [29]. ASO therapy, utilizing a short DNA sequence complementary to the *FMR1-217* isoform, was shown to bind and block this abnormal transcript. This intervention reduced *FMR1-217*, and thus got it (partially) out of the way, to rescue and restore normal mature *FMR1* RNA, and normalized FMRP to normal levels [29]. Interestingly, activation of the *FMR1* gene in transcriptionally silent FXS cells (replicating FXS individuals without any *FMR1* RNA) using the DNA demethylation agent 5-aza-2′-deoxycytidine (5-AzadC) increased *FMR1-217* RNA levels but failed to restore FMRP. Although pre-treatment with ASOs prior to 5-AzadC restored full-length *FMR1* expression and FMRP production [29]. These results suggest that *FMR1-217* RNA may be the predominant form of *FMR1* RNA and that its presence hinders the production of functional *FMR1* RNA and FMRP. This could explain why previous attempts to reactivate the *FMR1* gene via demethylation were unsuccessful.

Although promising, ASO therapy has thus far only been tested in vitro, and additional studies are required to validate these findings. Future research must confirm whether *FMR1* is not entirely silent in FXS, as previously thought, but instead transcribed into aberrant RNA that fails to produce FMRP. Progress in ASO therapy is limited by the scarcity of postmortem FXS brain tissue and the need for novel mouse models that allow mis-splicing of *FMR1* RNA without complete gene knockout. Nevertheless, these findings indicate that misregulated RNA processing events could serve as biomarkers for FXS and that ASO therapy targeting *FMR1-217* may provide a viable therapeutic approach for FXS individuals expressing this isoform.

### 7.6. Challenges of Brain-Targeted Gene Therapy in FXS

Delivering gene therapy components to the brain presents significant challenges. The blood–brain barrier restricts the delivery of therapeutic agents, including viral vectors, to affected brain regions. Effective and minimally invasive delivery methods are essential, as uneven distribution within brain tissue often results in incomplete targeting, particularly in large or deep brain structures that play a role in complex behaviors. Furthermore, the use of viral vectors is associated with potential risks, including immune responses, inflammation, toxicity, and off-target effects, which raise concerns about unintended gene expression in non-target tissues, potentially leading to harmful consequences. Most studies exploring gene therapy have been conducted in FXS animal models or cellular systems, which are valuable but lack complex neural circuitry found in humans and are thus potentially limited in their ability to fully recapitulate the pathophysiology of FXS.

Another critical challenge is achieving long-term, sustained, and regulated expression of the *FMR1* gene. Both overexpression and insufficient expression of *FMR1* may lead to adverse outcomes; thus, precise control over the level of gene expression is crucial. The timing of gene therapy intervention is also important. Earlier treatments, such as those administered in utero or during the neonatal period, may offer better outcomes but raise more ethical concerns, including the potential for unintended consequences. Administering gene therapy during early childhood, prior to the full onset of symptoms, may also improve outcomes. Although, in some cases, lifelong administration might be necessary to sustain adequate levels of FMRP. Ongoing research and clinical trials will be essential in refining therapeutic strategies to ensure both efficacy and ethical integrity in the context of gene therapy for FXS.

## 8. Bridging the Gap: Overcoming Challenges in Translating FXS Treatments

It is encouraging to see that several disrupted pathways in FXS appear to be amenable to pharmacological intervention, alongside emerging gene therapy strategies. Preclinical studies, particularly in the FXS mouse model, have shown promising results. However, translating these findings into effective clinical treatments for individuals with FXS remains challenging, as discussed in the following sections.

### 8.1. Preclinical Testing Protocols

A major challenge in preclinical studies is the inconsistency in behavioral test batteries and protocols across different studies and laboratories, which complicates reliable comparisons of drug efficacy. To address this issue, the development and implementation of standardized, time-efficient testing protocols are essential. Such systems would allow researchers across multiple sites to assess treatment effectiveness more reliably, enable data sharing, and ultimately enhance the translation of preclinical findings into clinical success. An example is the Live Mouse Tracker (LMT), a system that integrates radio-frequency identification (RFID), computer vision, and machine learning to monitor multiple mice in real-time [191]. The LMT enables the automatic annotation of individual and social behaviors over extended periods, facilitating a detailed analysis of interactions within groups. Notably, LMT can track up to four mice simultaneously, compared to standard behavioral tests that typically assess a single mouse. Furthermore, LMT enables continuous monitoring over 24 to 48 h, making it suitable for toxicity studies while capturing both day and night phases of behavior, whereas conventional tests are restricted to short observation periods.

Another critical limitation of most preclinical studies is the use of single acute doses, leaving the impact of long-term, chronic dosing largely unexplored. This raises concerns about potential treatment tolerance or diminishing efficacy over time. For instance, mGluR5 negative allosteric modulators exhibited treatment resistance in mice, yet long-term effects were not adequately explored before advancing to clinical trials that ultimately failed [192]. Addressing this gap by incorporating chronic dosing studies in preclinical research could better predict long-term treatment outcomes. Additionally, many studies have focused on treating adult mice after the onset of symptoms. Given that FXS is a neurodevelopmental disorder with profound effects on brain development, earlier intervention may be more effective in mitigating disease progression. Initiating treatment at an earlier stage could better model the developmental impact of FXS and enhance therapeutic success.

### 8.2. Clinical Trial Designs

Improvements are needed not only in preclinical research but also in the clinical trial process itself. It is essential to recognize that the successes observed in preclinical evaluations have rarely been replicated in clinical trials. Clinical trial design is crucial, particularly in selecting appropriate outcome measures for FXS. Common challenges included underpowered cohorts, short intervention durations, patient heterogeneity, and limited studies in younger children who may have greater benefits from early interventions. To maximize potential for therapeutic breakthroughs, clinical trials should aim to include individuals with the full FXS mutation, ensuring that those most likely to benefit are adequately represented.

The assessment of treatment efficacy is challenging, as FXS individuals often score at the lower end of standard cognitive tests, making it difficult to assess specific treatment effects [193]. Other studies relied on caregiver-reported primary outcome measures, susceptible to placebo effects and lacking in direct assessments of cognition and functional abilities. Recently, significant progress has been made in developing and validating FXS-specific outcome measures. These refined tools are designed to facilitate a more precise interpretation of clinical data and are able to detect subtle therapeutic responses in subgroups of FXS individuals [193,194,195]. Implementing these measures in clinical trials could substantially improve the evaluation of new treatments.

The heterogeneous nature of FXS complicates the selection of consistent outcome measures, making it challenging to predict treatment success across preclinical and clinical studies. A critical need exists for a biomarker that is observable in both animal models and humans, responsive to intervention, and predictive of clinical outcomes. One promising biomarker could be to measure electrophysiological changes in the brain, using EEG profiles. As reviewed by Kenny et al., EEG differences observed in FXS individuals show similarities to those found in FXS animal models, which support its potential as a translational biomarker [196]. A recent study demonstrated that resting-state EEG can reliably capture reproducible measures of brain activity in individuals with FXS, making it a valuable objective marker for assessing treatment effects [197]. Efforts are also underway to simplify and adapt EEG protocols for in-home use, which will be helpful for future clinical trials.

Another promising biomarker involves measuring trace amounts of FMRP in dried blood spots from individuals with FXS. This method has shown a positive correlation between FMRP levels and IQ scores in both males and females with FXS [198]. Such an approach offers a practical way to identify patient subgroups that are more likely to respond to specific pharmacological treatments, thereby improving the precision and efficacy of clinical trials. Importantly, pharmacological interventions should not be considered in isolation. Combining drug treatments with educational, therapeutic, and behavioral interventions can provide a comprehensive approach to optimizing functional outcomes for individuals with FXS. By refining clinical trial designs, incorporating robust biomarkers, and integrating multimodal therapeutic strategies, the field can move closer to developing effective, evidence-based treatments for FXS.

## 9. Conclusions

Significant progress has been made in understanding FXS over the past 80 years, yet individuals affected by the disorder continue to await effective therapies. Over the past few decades, extensive research has led to key advancements in the genetic, molecular, and clinical understanding of the disorder. The identification of the *FMR1* gene mutation in 1991 marked a pivotal breakthrough, establishing FXS as the most common inherited cause of intellectual disability and a leading monogenic contributor to autism spectrum disorder. The absence of FMRP disrupts critical neural pathways, affecting synaptic plasticity and contributing to cognitive, behavioral, and physical manifestation characteristics of FXS.

Animal models, particularly the *Fmr1* KO mouse, have played a crucial role in unraveling the pathophysiology of FXS. These models have been instrumental in identifying disrupted signaling pathways, such as mGluR5, GABAergic dysfunction, and many others. Despite providing valuable mechanistic insights, differences between animal and human neurodevelopment have posed challenges in direct clinical translation. Nevertheless, emerging therapeutic strategies, including AI-driven combination therapies, ASOs, and gene reactivation approaches, hold promise for addressing the underlying molecular deficits of FXS.

Recent clinical trials have targeted key disrupted pathways using pharmacological agents aimed at modulating synaptic function, neurotransmitter balance, and intracellular signaling cascades. The most promising candidates include PDE4D inhibitors, as well as metformin, and the anecdotal success stories with arbaclofen cannot be ignored. While some approaches have shown potential, variability in patient responses and the lack of robust outcome measures have posed challenges in achieving consistent therapeutic efficacy. A more personalized treatment approach, considering factors such as methylation status, FMRP expression levels, and comorbidities, may be necessary to optimize therapeutic outcomes.

Advances in gene-targeting strategies represent a paradigm shift in FXS research. Techniques such as AAV-mediated gene therapy, epigenetic editing, and ASO-based reactivation of the silenced *FMR1* gene offer the potential to address FXS at its root cause. However, these approaches face significant hurdles, including optimizing central nervous system delivery, ensuring long-term safety, and achieving precise control of gene expression. Additionally, combination therapies that simultaneously address multiple dysregulated pathways may offer a more comprehensive strategy for mitigating FXS symptoms.

Moving forward, several key areas will be critical in advancing FXS therapeutics. Standardized preclinical testing protocols, refined clinical trial designs, and identifying reliable biomarkers will be essential for assessing treatment efficacy. Furthermore, integrating behavioral and pharmacological interventions may enhance overall therapeutic outcomes by addressing both the biological and functional aspects of FXS. It is important to acknowledge that not all individuals with FXS require treatment, as many lead fulfilling lives with the support of their families or within assisted living environments. Future therapeutic strategies should therefore prioritize personalized and need-based interventions that respect individual well-being and quality of life.

Overall, FXS serves as a prime example of the challenges and opportunities in translating neurodevelopmental research into clinical applications. Despite setbacks in past clinical trials, continued advancements in targeted therapies, precision medicine, and gene-based interventions provide cautious optimism for future breakthroughs. With sustained research efforts and collaborative approaches, the development of effective personalized treatments for FXS remains an achievable goal, ultimately improving the quality of life for affected individuals and their families.

## Figures and Tables

**Figure 1 biomedicines-13-00805-f001:**
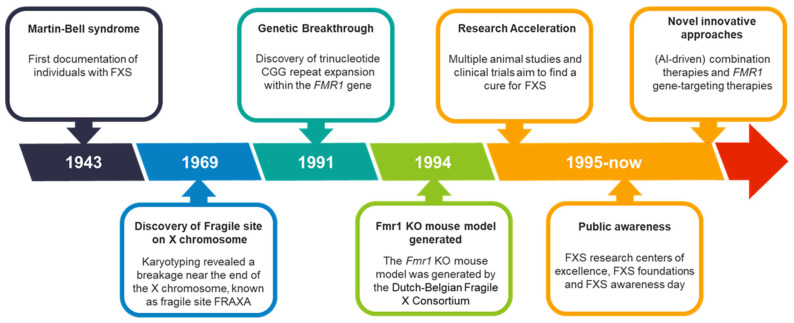
FXS over the past 80 years. FXS timeline with milestones, breakthroughs, and key events that have shaped the evolving understanding of the disorder over the past 80 years.

**Table 1 biomedicines-13-00805-t001:** Common features of FXS.

Phenotype	Symptoms
Cognitive Phenotype	Cognitive deficits
	Intellectual disabilities
	Developmental delays (motor and/or language)
Behavioral Phenotype	Hyperactivity
	(Social) anxiety
	Aggression
	Impulsivity
	Attention deficits
	Hyperarousal to sensory stimuli
	Repetitive stereotypic behaviors (hand flapping)
	Echolalia (unusual speaking patterns)
	Impaired social skills (ASD)
	Gaze avoidance
Physical Phenotype	Elongated face
	Large ears
	Prominent jaw and forehead
	High-arched palate
	Hyperflexible joints
	Flat feet
	Macroorchidism (large testicles in teens/adults)
Comorbid Health Issues	Recurrent otitis media (ear infection)
	Strabismus (crossed eyes)
	Weight gain
	Seizures
	Gastrointestinal problems
	Sleep problems

**Table 2 biomedicines-13-00805-t002:** Current actively recruiting interventional studies for FXS (ClinicalTrials.gov, March 2025).

Nct Number	Interventions	Sponsor
NCT06334419	Gaboxadol (GABA_A_ receptor agonist)	Children’s Hospital, Cincinnati
NCT05418049	Baclofen (GABA_B_ receptor agonist)	Children’s Hospital Cincinnati
NCT05418049	Memantine (NMDA antagonist)	Children’s Hospital Cincinnati
NCT05418049	Roflumilast (PDE4 inhibitor)	Children’s Hospital Cincinnati
NCT05367960	Zatolmilast/BPN14770 (PDE4D inhibitor)	Tetra Discovery Partners
NCT05358886	Zatolmilast/BPN14770 (PDE4D inhibitor)	Tetra Discovery Partners
NCT05163808	Zatolmilast/BPN14770 (PDE4D inhibitor)	Tetra Discovery Partners
NCT05120505	Metformin (activation of AMPK)	Children’s Hospital of Fudan University
NCT03862950	Metformin (activation of AMPK)	University of Alberta
NCT04977986	ZYN002 (modulate EC system)	Zynerba Pharmaceuticals, Inc.

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
