# Peer review of "From Discovery to Innovative Translational Approaches in 80 Years of Fragile X Syndrome Research"

_biomedicines, 2025, doi:10.3390/biomedicines13040805_

Round 1
Reviewer 1 Report
Comments and Suggestions for Authors
Please see attached file.

It is all stated in the attached file.
Author Response
Van der Lei and Kooy wrote a detailed review on the history, the clinical phenotypes, the pathophysiology, and the treatments of Fragile X Syndrome (FXS). The review is comprehensive, although a few references could be added:
- Genovese, A. C. and Butler, M. G. (2025). Systematic Review: Fragile X Syndrome Across the Lifespan with a Focus on Genetics, Neurodevelopmental, Behavioral and Psychiatric Associations. Genes 16, 149.: a very recent review on FXS
Response: Thank you, this reference has been added to the manuscript.
- Drozd et al. (2018). Modeling Fragile X Syndrome in Drosophila. Front. Mol. Neurosci. 11, 124 and Trajković et al.. (2022). Drosophila melanogaster as a Model to Study Fragile X-Associated Disorders. Genes 14, 87: two reviews on the Drosophila model, which is not much presented.
Response:.Thank you, both references have been included in the the manuscript.
In the Introduction, it would be nice to explain what is known or not known on the causes of the expansion of the CGG repeats and what are the functions of the KH and RGG protein domains. The non-canonical K0 domain could be put in the figure as its existence is now widely accepted.
Response: Thank you, we have expanded the discussion on CGG repeat expansion and included a reference to a recent review. The KH and RGG protein domains are now explained in the text. The K0 domain has been added to the figure as requested.
In table 2, Zatolmilast is an inhibitor of PDE4D not of PDE4. This table describes 12 clinical trials, although the site ClinicalTrials.gov states that 112 studies are currently ongoing. This has to be mentioned and to be explained why some drugs (ex. Ganaxolone) are not in the table.
Response: Thank you, we corrected the description of Zatolmilast to specify PDE4D inhibition. The discrepancy in trial numbers has been addressed: we conducted a targeted search filtering for FXS, interventional studies, and actively recruiting trials, which reduced the number of included studies to those listed in the table. Only currently recruiting trials (as of March 2025) are included, while past unsuccessful trials are omitted. The text on BK channel modulators has been updated to reflect the latest published results.
It is not reported the article showing that MPEP cures FXS in Drosophila (McBride et al., 2005. Pharmacological Rescue of Synaptic Plasticity, Courtship Behavior, and Mushroom Body Defects in a Drosophila Model of Fragile X Syndrome. Neuron 45, 753– 764.), which was published before the studies in mice.
Response: Thank you, this reference has now been included in the section discussing MPEP.
Line 468 should be corrected: metformin did not improve circadian rhythm. See page 1146 of Monyak et al. (2017): “Although we were unable to rescue circadian rhythmicity with any combination of developmental and adulthood metformin treatment (Supplementary Figures 5a and b), we found that…”
Response: Thank you, this error has been corrected and the claim removed.
When describing AI-driven therapies, it would be important to specify which software programs have been used.
Response: Thank you, additional details and references on Healx and Kantify AI-driven approaches have been included, although proprietary software remains unpublished.
The chapter 6.4 mentioning “recent advances in…” presents publications that are quite out-dated. What has been done since then? If nothing, “recent” should be eliminated.
Response: Thank you, The term "recent" has been removed from the section title.
The review is quite well written, but the following editing corrections are required:
- Line 245: from humans
- Line 289: delete “are”
- Line 360: a question mark is missing after “model”.
- Lines 450-452: the sentence starting with “Although” is not correct.
- Line 507: “writing this thesis” is inappropriate.
- Line 524: “target” instead of “targeting” or remove “that”
- Line 577: reactivate
- Line 599: functional
- Line 759: remove comma before “and”.
The comma before “and” should be put only when more than three words or groups of words precede it. Please check other cases and be consistent.
- Line 770:remove comma after ASO.
Response: Thank you for pointing out these errors, all corrections have been implemented in the manuscript.
Reviewer 2 Report
Comments and Suggestions for Authors
This review provides a comprehensive overview of Fragile X syndrome (FXS), encapsulating over 80 years of research spanning its clinical features, genetic mechanisms, animal models, disrupted molecular pathways, and therapeutic advancements. The discussion on translational challenges, including patient heterogeneity and the need for refined clinical trial methodologies, is particularly valuable. I read through this review and believe this review is well-structured and written in perfectly clear language. I have only several minor suggestions to revise.
Minors
Line 66-67: The statement regarding the normal CGG repeat range (5 to 55) and pathological expansion (>200) lacks proper citations. To ensure accuracy and traceability, references should be provided to support these figures.
Line 86-87: The discussion on AI-driven combination therapies would benefit from additional elaboration. In specific, a reference supporting the rationale for AI-based approaches in targeting multiple molecular pathways in FXS would strengthen this section. AI applications in FXS treatment are an emerging area of research and should be contextualized with relevant studies.
Figure 1: Each milestone presented in the figure should be accompanied by appropriate citations to enhance traceability and allow readers to verify key developments in FXS research.
Figure 2: The current diagram quality is suboptimal. A vector-based image format or a high-resolution JPEG should be used to improve clarity and readability.
Lines 263-269: The discussion on disrupted signaling pathways, including ERK, mTOR, and MMP families, requires citations to original studies. Providing references for each signaling pathway will substantiate the claims and ensure the accuracy of the review.
Author Response
This review provides a comprehensive overview of Fragile X syndrome (FXS), encapsulating over 80 years of research spanning its clinical features, genetic mechanisms, animal models, disrupted molecular pathways, and therapeutic advancements. The discussion on translational challenges, including patient heterogeneity and the need for refined clinical trial methodologies, is particularly valuable. I read through this review and believe this review is well-structured and written in perfectly clear language. I have only several minor suggestions to revise.
Line 66-67: The statement regarding the normal CGG repeat range (5 to 55) and pathological expansion (>200) lacks proper citations. To ensure accuracy and traceability, references should be provided to support these figures.
Response: Thank you for bringing this to our attention. We have now added appropriate citations to support the stated CGG repeat ranges and ensure accuracy.
Line 86-87: The discussion on AI-driven combination therapies would benefit from additional elaboration. In specific, a reference supporting the rationale for AI-based approaches in targeting multiple molecular pathways in FXS would strengthen this section. AI applications in FXS treatment are an emerging area of research and should be contextualized with relevant studies.
Response: We appreciate this suggestion. To provide a more comprehensive discussion, we have incorporated additional references in the introduction to establish the rationale for AI-based approaches in FXS treatment. Furthermore, we have revised the corresponding section to offer more context on AI-driven combination therapies, including their potential for targeting multiple molecular pathways.
Figure 1: Each milestone presented in the figure should be accompanied by appropriate citations to enhance traceability and allow readers to verify key developments in FXS research.
Response: Thank you, we have now added citations to all key milestones in Figure 1 to enhance traceability and ensure that readers can verify these developments.
Figure 2: The current diagram quality is suboptimal. A vector-based image format or a high-resolution JPEG should be used to improve clarity and readability.
Response: Thank you, to improve clarity and readability, we have replaced Figure 2 with a high-resolution JPEG image.
Lines 263-269: The discussion on disrupted signaling pathways, including ERK, mTOR, and MMP families, requires citations to original studies. Providing references for each signaling pathway will substantiate the claims and ensure the accuracy of the review.
Response: Thank you, we have now added relevant citations to support each of the discussed signaling pathways.
Reviewer 3 Report
Comments and Suggestions for Authors
Thank you for inviting me to write this very well-written and engaging review of 80 years of research into Fragile X Syndrome.
I leave it to the discretion of the editors whether they wish to impose some additional structure by asking the authors to define the type of review they did (scoping, topical, historical, etc.) and apply the relevant rules of that particular review type (i.e., PRISMA diagram, etc.).
I also recommend this edit for lines 679-683: "However, translating these findings into effective clinical treatments for individuals with FXS remains challenging, as discussed in the following sections." (The language of "double edged sword" applies better to the previous section where you discuss the unforeseen consequences/"off-target effects" of genetic therapies).
Author Response
Thank you for inviting me to write this very well-written and engaging review of 80 years of research into Fragile X Syndrome.
I leave it to the discretion of the editors whether they wish to impose some additional structure by asking the authors to define the type of review they did (scoping, topical, historical, etc.) and apply the relevant rules of that particular review type (i.e., PRISMA diagram, etc.).
Response: up to the editors
I also recommend this edit for lines 679-683: "However, translating these findings into effective clinical treatments for individuals with FXS remains challenging, as discussed in the following sections." (The language of "double edged sword" applies better to the previous section where you discuss the unforeseen consequences/"off-target effects" of genetic therapies).
Response: thank you, we have edited these lines as you suggested.